# ELDET: Early-Learning Distillation with Noisy Labels for Object Detection

**Dongmin Choi**[1*]  **Sangbin Lee**[1,2*]  **EungGu Yun**[1]  **Jonghyuk Baek**[3†]  **Frank C. Park**[1,2]

[1]SAIGE  [2]Seoul National University  [3]Flitto

{dm.choi, sb.lee, eg.yun}@saige.ai, jonghyuk.baek@flitto.com, fcp@snu.ac.kr

## Abstract

The performance of learning-based object detection algorithms, which attempt to both classify and locate objects within images, is determined largely by the quality of the annotated dataset used for training. Two types of labelling errors are prevalent: objects that are incorrectly classified (*categorization noise*) and inaccurate bounding boxes (*localization noise*); both errors typically occur together in large-scale datasets. In this paper, we propose a distillation-based method to train object detectors that takes into account both categorization and localization noise. The key insight underpinning our method is that the early-learning phenomenon – in which models trained on noisy data with mixed clean and noisy labels tend to first fit to the clean data, and memorize the noisy labels later – manifests earlier for localization noise than for categorization noise. We propose a method that uses models from the early-learning phase (before overfitting to noisy data occurs) as a teacher network. A plug-in module implementation compatible with general object detection architectures is developed, and its performance is validated against the state-of-the-art using PASCAL VOC, MS COCO and VinDr-CXR medical detection datasets.

## 1 Introduction

Object detection is a fundamental task in computer vision that requires both accurate classification and precise localization of objects within images. Object detection is essential for autonomous driving [12, 54, 53], medical imaging [20, 58, 37] and many other applications that rely on knowing both the type and location of objects in images.

With few exceptions, methods for object detection are now almost entirely based on neural network models, *e.g.*, [43, 46, 35, 6, 44, 26] trained on large image datasets such as PASCAL VOC [11] and MS COCO [30]. While network architectures and algorithms for data processing and training

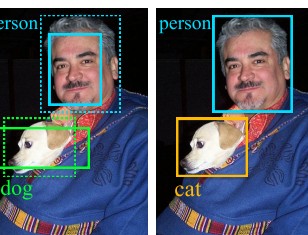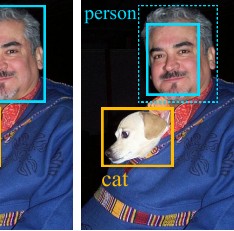

(a) Localization noise  (b) Categorization noise  (c) Combined noise

Figure 1: Comparison of label noise types in object detection: (a) Localization only. (b) Categorization only. (c) Combined localization and categorization noise.

may still affect object detection performance, by and large the performance of object detectors is determined mostly by the size and quality of the datasets [38].

---

[*]These authors contributed equally to this work.

[†]This work was primarily conducted while the author was at SAIGE.

39th Conference on Neural Information Processing Systems (NeurIPS 2025).

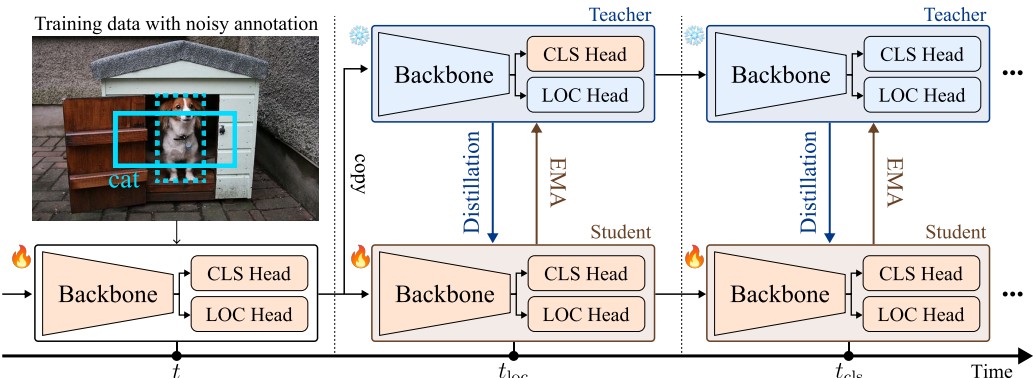

Figure 2: Overview of the ELDET workflow. The early-stage model is initially trained on a dataset with noisy labels until memorization occurs at the $t_{loc}$-th training iteration. Once the early-learning phase terminates for the localization task, a teacher model (blue), initialized from the current state of the student model (orange), is employed to guide subsequent training through knowledge distillation with its parameters frozen. The teacher model is progressively updated using an exponential moving average (EMA) of the student parameters after each iteration. Concurrently, the classification head is actively updated with a small momentum prior to the memorization phase for classification at iteration $t_{cls}$.

Acquiring high-quality annotations for large-scale datasets is difficult. Even with recently available methods for auto-labelling and label assist [64, 10], some manual effort is unavoidable, and human labeling errors are often the main source of noisy labels [55, 52]. The two main types of labeling errors are misclassification (*categorization noise*) and inaccurate bounding boxes (*localization noise*). In large-scale, crowd-sourced annotations [13, 63] both errors frequently occur together; for instance, labeling errors of the type shown in Figure 1 are quite common. As observed in [31, 3], the performance of object detection models that are overfit to such incorrect labels can be significantly degraded.

For the simpler task of image classification, numerous methods address noisy labels in [32, 34, 23, 25, 2, 27]. In comparison, only a few works address noisy labels in object detection, with the focus typically on a single type of noise (categorization or localization) rather than both simultaneously. For example, Liu *et al.* [36] propose a method to mitigate categorization noise by excluding unreliable samples, while Bär *et al.* [3] introduce a localization label refinement network to correct box errors. These approaches are limited in real-world settings where both types of noise often appear together.

In this paper, we propose a method to train object detectors that takes into account both categorization and localization noise in the annotations. The key insight underpinning our method is the *early-learning phenomenon* [32] – in which models trained on noisy data with mixed clean and false labels have been observed to first fit to the clean data, and that the false labels are memorized later – manifests differently for classification and localization errors. Specifically, we observe that *models tend to memorize localization error earlier than categorization noise*.

Leveraging this observation, we propose an object detection training algorithm based on knowledge distillation [19, 50], in which models from the early-learning phase (before overfitting occurs to noisy data) act as teacher networks. Specifically, once the early-learning phase terminates, we copy the model to create a frozen teacher network, while a student model continues training on the noisy data. To detect the transition from early-learning to memorization, we fit an exponential parametric curve to training metrics and identify the transition point when the curve's gradient significantly decreases [31, 33]. The teacher network is updated by an exponential moving average (EMA) of the student's parameters to train informative features from clean samples while maintaining robust knowledge. We further adjust EMA momentum for the classification and localization heads to handle the distinct early-learning termination points between these tasks effectively. This self-distillation workflow [21] helps the current model focus on learning from clean data while avoiding overfitting to noisy labels. The overview is illustrated in Figure 2.

We develop a practical implementation of our proposed method in the form of a plug-in module, compatible with various object detection architectures including both anchor-based [46, 44] and anchor-free models [49]. Our approach does not require architecture-specific modifications, making it widely applicable. We evaluate our approach on PASCAL VOC and MS COCO with noise

simulations. Additionally, we assess the domain robustness of our framework on the VinDr-CXR medical detection dataset [39], demonstrating its adaptability to specialized domains. Extensive experimental results demonstrate that our method significantly improves detection performance in noisy environments, effectively handling simultaneous classification and localization noise in real-world object detection tasks.

## 2 Related Works

**Object Detection.**   Object detection has primarily evolved along two directions: anchor-based [35, 65, 43] and anchor-free detectors [22, 66, 6]. Anchor-based detectors such as RetinaNet [46] and Faster R-CNN [44] use pre-defined anchor boxes as reference points to predict objects. In contrast, anchor-free detectors like FCOS [49] remove the dependency on anchor boxes by directly predicting object locations, which results in simpler architectures and reduced computational demands.

These methods can also be categorized into one-stage and two-stage detectors. One-stage detectors (*e.g.,* RetinaNet [46] and Generalized Focal Loss (GFL) [26]) aim to directly predict class probabilities and bounding box coordinates from the entire image in a single shot. Two-stage detectors (*e.g.,* Faster R-CNN [44] and Cascaded R-CNN [5]) first generate region proposals and then refine them for accurate prediction. Given these significant architectural differences, methods that can effectively integrate with both structures are clearly needed [57]. To this aim, our proposed approach is designed as a plug-in module that can be readily incorporated into most existing architectures.

**Transition from Early Learning to Memorization Under Noisy Annotations.**   Liu *et al.* [32] observe the following *early-learning phenomenon* in deep learning models: during the initial stages of training, model gradients are dominated by clean labels, while memorization of noisy labels emerges in later stages. Based on this observation, they propose a regularization technique that leverages the predictions of earlier models to limit the impact of noisy labels.

Building on this foundation, several works have aimed to develop robust models under noisy annotations for image classification [34, 25, 2, 27]. Han *et al.* [15] introduce Co-teaching, in which two networks are trained simultaneously using the small-loss instances selected by its peer network. Li *et al.* [23] propose DivideMix, which models the distribution of losses to separate clean and noisy samples, then applies semi-supervised learning techniques to utilize both sets effectively.

Extending the concept to segmentation, Liu *et al.* [33] introduce an Adaptive Early Learning Correction (ADELE) framework for the weakly-supervised setting. ADELE monitors class-specific transitions from early-learning to memorization, refining noisy labels with pseudo-labels generated from early-phase model predictions for each class. While segmentation and object detection both require localization and classification, ADELE's focus on pixel-wise annotations and class-specific early learning is more suited to segmentation tasks where localization involves fine-grained pixel boundaries. In contrast, object detection involves both the accurate localization of bounding boxes and the classification of entire regions within the box, making it challenging to directly apply ADELE to this domain. Thus, when applying early learning to object detection with noisy labels, what is needed is a specialized approach that addresses the unique challenges of noisy labels.

**Object Detection with Noisy Annotations.**   Some recent works that propose to train robust object detectors under noisy labels include [48, 24, 7]. Liu *et al.* [36] propose an adaptive framework which identifies reliable examples in noisy data by measuring instance-level domain properties and adjusting the training accordingly. While their approach effectively mitigates the impact of domain shifts with noise, it focuses primarily on domain adaptation scenarios and may not fully address the challenges posed by noisy labels in general object detection tasks.

ORSOD [31] introduces a dynamic loss decay mechanism to enhance the robustness of oriented object detection with noisy labels by adaptively reducing the influence of high-loss samples. However, ORSOD primarily addresses categorization noise and does not account for localization noise, which can significantly impact detection accuracy in real-world scenarios where both types of noise are present. In contrast, Bär *et al.* [3] propose a Localization Label Refinement Network (LLRN) to refine the noisy coordinates of bounding boxes. LLRN focuses on correcting localization errors by training a separate network to predict more accurate bounding boxes, which are then used to update the training data. While effective for localization noise, LLRN overlooks categorization noise,

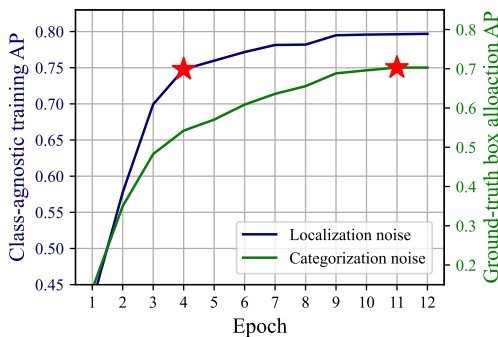

Figure 3: Evaluation results on training samples under localization (blue) or categorization noise (green) with class-agnostic and ground-truth box allocation metric, respectively. Red stars mark early-learning termination points for each metric of RetinaNet on PASCAL VOC with a noise ratio of 40%.

which presents clear limitations in real-world scenarios. In contrast to prior works, our approach simultaneously tackles both types of noise.

**Knowledge Distillation.** Knowledge distillation has been widely adopted to transfer informative features from a high-performance model (teacher) to an efficient network (student) [19]. Several works have proposed a distillation framework for object detection that improves the performance of small detectors [8, 51, 59]. Wang *et al.* [50] propose an efficient approach named CrossKD, which propagates the intermediate features of the student network to the heads of the teacher models. This cross-head approach enables task-oriented knowledge transfer between networks, improving the student's performance without significantly increasing computational costs.

In self-knowledge distillation [14, 56, 61, 21, 59, 28], a model serves as both teacher and student, gradually refining knowledge by jointly learning from the ground truth and past predictions of the model itself. Works adopting the co-teaching framework [15, 4, 29] demonstrate that self-knowledge distillation can address noisy labels by distilling knowledge from the model itself. While these works have shown promising results, they often suffer from high computational costs due to training multiple models simultaneously.

## 3 Priority in Memorization: Localization Over Classification

The dual-task nature of object detection—classification and localization—lead us to hypothesize that detectors may prioritize the coupled objectives differently when training under noisy labels. To investigate this phenomenon, we conduct empirical experiments to monitor how object detectors internalize noisy labels across the two tasks.

Specifically, we adopt task-specific metrics to independently monitor detector performance while isolating the impact of noise on each task. First, we evaluate detectors in a class-agnostic (CA) manner, treating all objects as a single category. This evaluation prevents potential interference from classification errors. For categorization noise, we replace the predicted bounding box coordinates with the corresponding ground truth if the overlap between the two exceeds a certain threshold. This ground-truth box allocation (GTBA) effectively alleviates the influence of localization errors, allowing for a more accurate assessment of classification performance. Details are discussed in Section 4.1.

As illustrated in Figure 3, our findings reveal a critical aspect of model behavior under noisy labels for object detection. We observe that *the memorization of localization noise occurs significantly earlier than the memorization of categorization noise*. This finding suggests that the model tends to focus on spatial aspects during the early learning phase, while requiring longer training times for classification. Based on this finding, we propose a technique that suppresses noise memorization across the two tasks.

# 4 Methods

In this section, we discuss our approach for training robust object detectors in the presence of noisy labels. We exploit the early learning phenomenon [32] to address the challenges posed by both localization and categorization noise.

## 4.1 Early-Learning and Memorization

As discussed in Section 3, the early learning phenomenon manifests differently in object detection for the two key tasks of localization and classification. Specifically, models tend to memorize localization noise earlier than categorization noise. To accurately capture any task-specific patterns, we employ specialized metrics that independently monitor the training progress of each task.

For localization, we utilize a class-agnostic (CA) metric that treats all detected objects as belonging to a single category (e.g., "object"). This evaluation isolates localization performance without the confounding influence of classification errors, allowing us to monitor how the model learns spatial information over time. For classification, we introduce a Ground Truth Box Allocation (GTBA) approach. Inspired by Moon *et al.* [60], we replace the box coordinates of the model prediction with the ground truth box if the Intersection over Union (IoU) between them exceeds a certain threshold $\tau$. This allocation effectively eliminates the impact of localization errors, enabling a more accurate assessment of classification performance under noisy label conditions.

By employing these task-specific metrics, we can effectively discern the distinct dynamics of early-learning across localization and classification tasks, and determine the appropriate moments to intervene during training.

## 4.2 Early Learning Phase Detection

To detect the transition from early learning to memorization for each task, we follow Liu et al. [33] in fitting an exponential parametric function to model the rate of change with respect to the performance on noisy training sets over training:

$$f(t) = a \left( 1 - e^{-b \cdot t^c} \right), \tag{1}$$

where $t$ represents the training epoch, and $a > 0$, $b \geq 0$, and $c \geq 0$ are parameters that are fitted to the observed data. This parametric form captures the performance trend of the model on the training dataset.

To detect the transition point from early-learning to memorization, we monitor the relative change in the derivative of the metric. Specifically, the transition is identified when the rate of change deviates significantly from the initial learning gradient, as formalized by the following condition:

$$\frac{|f'(1) - f'(t)|}{|f'(1)|} > \gamma, \tag{2}$$

where $\gamma$ is a threshold capturing the deviation from the initial learning rate. When this condition is met, it indicates that the model dynamics have shifted and it begins to memorize noisy annotations. Using this method, we determine the transition epochs $t_{\text{loc}}$ and $t_{\text{cls}}$ for localization and classification respectively, based on the metrics in Section 4.1.

## 4.3 Early Learning Guidance via Distillation

**Knowledge Distillation.** To mitigate the effects of noisy labels and prevent memorization, we propose an early learning guidance mechanism. This mechanism involves using the model obtained at the end of the early learning phase as a teacher network in a knowledge distillation framework [19, 50, 21]. The teacher network, which has not yet overfitted to noisy labels, guides the current student model during subsequent training, helping it avoid overfitting to noise. The student model continues training on the noisy dataset but receives guidance from the teacher model to focus on learning from clean data. The distillation process is implemented by minimizing the divergence between the predictions of the teacher model and the student model. Specifically, we define the knowledge distillation loss:

$$\mathcal{L}_{\text{kd}} = \mathcal{L}_{\text{cls}}^{\text{kd}} + \mathcal{L}_{\text{loc}}^{\text{kd}}, \tag{3}$$

where $\mathcal{L}_{\mathrm{cls}}^{\mathrm{kd}}$ and $\mathcal{L}_{\mathrm{loc}}^{\mathrm{kd}}$ are the distillation losses for classification and localization, respectively. The total loss function for training the student model is then defined as

$$\mathcal{L}_{\mathrm{total}} = \mathcal{L}_{\mathrm{det}} + \lambda\mathcal{L}_{\mathrm{kd}}, \tag{4}$$

where $\mathcal{L}_{\mathrm{det}}$ is the original detection loss between the student predictions and the noisy ground-truth annotations, and $\lambda$ is a hyperparameter that balances the detection loss and the distillation loss.

To enhance the efficiency of the distillation process, we adopt the CrossKD framework [50]. CrossKD allows for knowledge transfer between the teacher and student models by propagating the intermediate features of the student network to the teacher's detection heads. This cross-head approach enables task-oriented knowledge transfer without significant computational overhead.

**Teacher Model Update via EMA.** To ensure that the teacher model remains a reliable source of guidance throughout training, co-teaching frameworks [15, 29] that propagate gradients to both teacher and student networks can be used. However, this dual-training approach incurs high computational costs, as it requires training two models simultaneously. Instead, we update the teacher model using an exponential moving average (EMA) of the student model's weights without updating it directly. The EMA update rule is defined as follows:

$$\theta_{\mathrm{teacher}} \leftarrow \alpha\theta_{\mathrm{teacher}} + (1-\alpha)\theta_{\mathrm{student}}, \tag{5}$$

where $\theta_{\mathrm{teacher}}$ and $\theta_{\mathrm{student}}$ represent the weights of the teacher and student models, respectively, and $\alpha$ is the decay rate. Typically $\alpha$ is set close to 1 to ensure that the teacher model updates slowly, preserving its robustness to noisy labels. However, since the teacher model is initialized after the early-learning phase ends for localization but before it concludes for classification, using a high decay rate on the classification head can potentially hinder its performance. To preclude this possibility, we apply a small decay rate of $\alpha = 0.1$ specifically to the classification head during this period. This enables faster updates for the classification head, which allows classification performance to sufficiently converge before memorization occurs without compromising noise robustness.

Our framework combines EMA updates with knowledge distillation to enable effective teacher guidance and robust training, mitigating noisy label effects in object detection.

## 5 Experiments

### 5.1 Experimental Settings

**Noise Simulations.** To evaluate the robustness of object detectors under noisy labels, we simulate both localization and categorization noise. For localization noise, we perturb the ground truth bounding box coordinate while keeping its category [3]. Given a ground truth box $\mathbf{b} = (x_{\min}, y_{\min}, x_{\max}, y_{\max})$, the noisy box $\mathbf{b}^{\mathrm{noisy}}$ is calculated as:

$$\mathbf{b}^{\mathrm{noisy}} = \begin{bmatrix} x_{\min} + \delta_x & y_{\min} + \delta_y & x_{\max} + \delta_x' & y_{\max} + \delta_y' \end{bmatrix}^{\top}. \tag{6}$$

where $\delta_x, \delta_y, \delta_x', \delta_y'$ are sampled from uniform distributions scaled by box size with magnitude controlled by $\epsilon = 0.5$, corresponding to up to 50% perturbation relative to width or height.

For categorization noise, we randomly replace the true class label $c_i$ of each object with an incorrect label $c_i'$ drawn uniformly from all other possible classes $\mathcal{C} \setminus \{c_i\}$:

$$c_i' \sim \mathcal{U}(\mathcal{C} \setminus \{c_i\}). \tag{7}$$

We apply these noise simulations to a random subset of the training data with various levels of 20%, 30%, and 40%.

**Datasets.** We conduct our experiments on PASCAL VOC [11] and MS COCO, which is a widely used benchmark for object detection. Following the standard protocol [49], we use the VOC 2007 and VOC 2012 `trainval` sets (16,551 images) for training, and perform evaluation on the VOC 2007 `test` set (4,952 images). MS COCO is a large-scale dataset with 80 object categories, featuring over 330K images and more than 2.5 million labeled instances. We use the COCO 2017 version, training on the `train` split (118K images) and evaluating on the `val` split (5K images).

To further validate the robustness of our approach, we run experiments on VinDr-CXR medical object detection dataset [40]. VinDr-CXR dataset consists of 15,000 chest X-ray images annotated with 14 classes, which represent various thoracic disease findings and abnormalities. For all datasets, we apply the aforementioned noise simulations only to the training data while keeping the validation and test sets with clean annotations.

Table 1: Comparison of performance under noise annotations on Pascal VOC reported as AP@50. The results are shown for localization, categorization and combined noise with noise levels at 20%, 30%, and 40% as well as the clean setting. For each detector, the best AP scores are highlighted in bold, with the second-best scores underlined.

| Detector | Method | Clean | Localization Noise | | | Categorization Noise | | | Combined Noise | | |
|---|---|---|---|---|---|---|---|---|---|---|---|
| | | | 20% | 30% | 40% | 20% | 30% | 40% | 20% | 30% | 40% |
| RetinaNet | - | 75.07 | 74.00 | 73.83 | 73.13 | 70.67 | 69.03 | 67.43 | 70.27 | 68.07 | 65.63 |
| | ORSOD [31] | 75.41 | 74.17 | 73.97 | 73.43 | 71.13 | 68.90 | 67.33 | 70.37 | 66.57 | 65.47 |
| | ADELE [33] | 74.69 | 73.67 | 74.10 | 71.03 | 71.03 | 69.23 | 67.07 | 70.13 | 68.20 | 65.87 |
| | ELDET (ours) | 76.52 | 76.23 | 76.30 | 74.80 | 73.66 | 73.71 | 68.21 | 74.53 | 73.67 | 68.82 |
| FCOS | - | 72.23 | 71.00 | 70.67 | 70.60 | 67.37 | 64.97 | 62.63 | 66.57 | 63.33 | 60.13 |
| | ORSOD [31] | 71.63 | 68.99 | 70.99 | 70.94 | 67.55 | 64.47 | 62.80 | 67.36 | 63.06 | 60.51 |
| | ADELE [33] | 71.59 | 71.20 | 70.70 | 70.57 | 67.53 | 65.40 | 62.67 | 66.73 | 63.87 | 60.70 |
| | ELDET (ours) | 72.00 | 73.40 | 72.80 | 74.10 | 68.43 | 66.13 | 63.73 | 68.67 | 65.03 | 62.43 |
| Faster R-CNN | - | 73.89 | 69.48 | 69.49 | 69.25 | 66.95 | 65.15 | 63.65 | 66.68 | 64.84 | 62.02 |
| | ORSOD [31] | 70.77 | 68.91 | 68.94 | 68.88 | 66.40 | 64.84 | 63.40 | 65.85 | 64.24 | 63.01 |
| | ADELE [33] | 71.48 | 69.15 | 68.54 | 68.91 | 67.76 | 65.65 | 63.59 | 68.83 | 65.28 | 62.61 |
| | ELDET (ours) | 73.34 | 71.92 | 71.81 | 70.12 | 69.40 | 67.60 | 66.33 | 69.09 | 66.83 | 64.28 |
| GFL | - | 73.00 | 73.23 | 71.68 | 71.50 | 69.54 | 66.18 | 62.77 | 67.70 | 64.69 | 56.30 |
| | ORSOD [31] | 74.11 | 73.87 | 73.12 | 72.51 | 69.92 | 67.71 | 63.64 | 68.49 | 64.73 | 61.28 |
| | ADELE [33] | 73.05 | 73.82 | 73.46 | 72.29 | 69.14 | 67.49 | 64.19 | 68.41 | 66.04 | 62.71 |
| | ELDET (ours) | 75.28 | 75.23 | 74.57 | 74.44 | 69.68 | 67.57 | 65.35 | 69.82 | 66.77 | 63.57 |

Table 2: Comparison of performance under noise annotations on MS COCO val2017, reported as AP@50. The results are shown for localization, categorization and combined noise with noise levels at 20%, 30%, and 40% as well as the clean setting. The best AP scores are highlighted in bold.

| Detector | Method | Clean | Localization Noise | | | Categorization Noise | | | Combined Noise | | |
|---|---|---|---|---|---|---|---|---|---|---|---|
| | | | 20% | 30% | 40% | 20% | 30% | 40% | 20% | 30% | 40% |
| RetinaNet | - | 44.41 | 43.97 | 42.61 | 43.12 | 41.86 | 40.66 | 39.90 | 42.61 | 42.33 | 41.95 |
| | ELDET (ours) | 45.95 | 44.96 | 44.87 | 44.74 | 43.51 | 41.98 | 40.31 | 44.51 | 43.23 | 42.99 |
| FCOS | - | 44.02 | 44.34 | 44.36 | 43.11 | 41.91 | 41.73 | 39.26 | 43.16 | 42.66 | 42.13 |
| | ELDET (ours) | 45.89 | 45.17 | 44.94 | 44.83 | 43.36 | 42.66 | 41.02 | 44.39 | 43.70 | 43.59 |
| Faster R-CNN | - | 43.55 | 42.80 | 42.86 | 42.48 | 40.36 | 39.26 | 37.21 | 40.91 | 40.83 | 39.57 |
| | ELDET (ours) | 44.79 | 44.00 | 43.51 | 43.49 | 41.42 | 39.47 | 38.20 | 42.72 | 42.06 | 40.30 |
| GFL | - | 47.24 | 47.19 | 46.08 | 45.17 | 45.32 | 44.53 | 43.31 | 46.30 | 45.58 | 44.81 |
| | ELDET (ours) | 49.56 | 48.77 | 47.43 | 46.64 | 47.05 | 45.59 | 44.78 | 47.47 | 46.93 | 45.58 |

**Baselines.** We compare ELDET with two baselines:

- **ORSOD** [31]: ORSOD addresses categorization noise by dynamically excluding samples with high classification loss during training, which prevents the model to memorize noisy samples with large loss.

- **ADELE** [33]: We extend the role of ADELE to object detection under noisy annotations. Specifically, detectors begin to be supervised by the pseudo-label from the early-models instead of raw noisy labels.

## 5.2 Experimental Results

We use the mean Average Precision at an Intersection over Union (IoU) threshold of 0.5 (AP@50) for PASCAL VOC and COCO, and 0.4 (AP@40) for VinDr-CXR. To validate the compatibility of our method with various architectures, we conducted experiments with different detectors including RetinaNet [46], FCOS [49], Faster R-CNN [44] and GFL [26].

**PASCAL VOC.** Table 1 presents a quantitative comparison of ELDET against baseline methods on PASCAL VOC. ELDET consistently outperforms the comparison models across all scenarios. Notably, ELDET using the RetinaNet significantly surpasses all existing baselines with a large gap under the combined noise condition with a 40% noise level. This substantial improvement

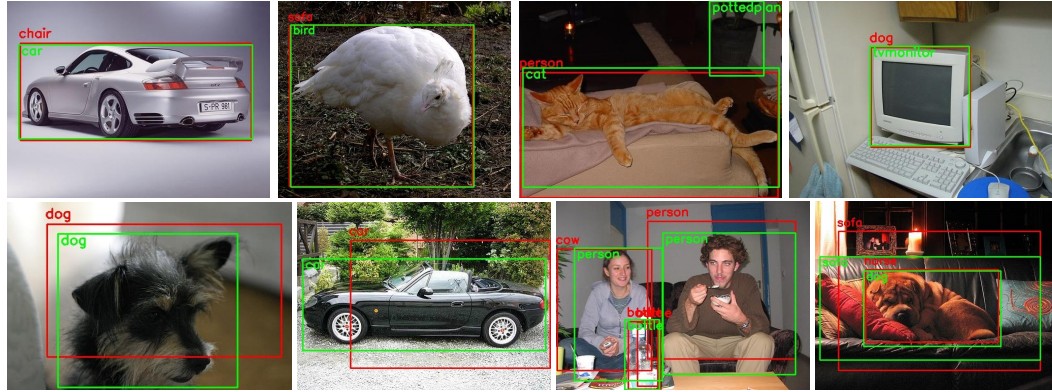

Figure 4: Qualitative results of ELDET on the training set of PASCAL VOC, where green boxes represent the predictions of our method and red boxes indicate the noisy annotations. Our model effectively avoids memorizing both localization and categorization noise.

Table 3: Comparison of detection performance under various noise levels on VinDr-CXR. Best AP scores are highlighted in bold.

| Detector | ELDET | Combined Noise | | |
|---|---|---|---|---|
| | | 20% | 30% | 40% |
| RetinaNet | – | 29.54 | 27.44 | 26.75 |
| | ✓ | **31.09** | **29.61** | **31.06** |
| FCOS | – | 28.51 | 27.09 | 25.32 |
| | ✓ | **34.40** | **32.55** | **32.74** |
| Faster R-CNN | – | 30.97 | 29.16 | 27.87 |
| | ✓ | **32.91** | **32.52** | **31.61** |
| GFL | – | 29.06 | 26.76 | 26.65 |
| | ✓ | **36.29** | **33.90** | **27.60** |

Table 4: Ablation study of task-specific metrics on PASCAL VOC with noise ratio of 40% using RetinaNet.

| | Class-agnostic | Ground-truth Box Allocation | AP |
|---|---|---|---|
| (1) | – | – | 66.13 |
| (2) | ✓ | – | 67.93 |
| (3) | – | ✓ | 65.70 |
| (4) | ✓ | ✓ | **68.82** |

underscores superior capability of ELDET to simultaneously handle both types of noise. Qualitative results are illustrated in Figure 4

In contrast, ORSOD and ADELE exhibit limited effectiveness in managing both types of noise. According to the original experimental results of ORSOD, it demonstrate negligible performance gain when integrated with certain architectures (*e.g.*, ReDet [16]), which indicates the low robustness of dynamic loss decay across different detectors. On the other hand, ADELE, which is initially designed for weakly-supervised settings where the ground truth mask quality is notably low, corrects the ground truth with early-learning phase predictions when memorization begins. However, this substitution can occur performance drop in our setting because direct supervision from early-phase predictions may prevent the model from effectively learning informative features.

**COCO.** Table 2 shows that incorporating ELDET consistently improves AP on the MS COCO dataset compared to the baseline without it. These gains on a large-scale benchmark underscore the robustness of our method and demonstrate its ability to generalize to complex, real-world data.

**VinDr-CXR.** As shown in Table 3, ELDET maintains its superior performance over the baselines on VinDr-CXR dataset. It is notable that ELDET boosts the performance of FCOS from 25.32 to 32.74 under a 40% combined noise condition. It shows the robustness and adaptability of our proposed method in challenging settings with complex noise patterns in medical imaging. By effectively handling noisy annotations in multi-domain, ELDET showcases its potential for widespread scenarios across various fields where annotation noise is prevalent to happen.

**Ablation Study on Task-Specific Early Learning.** We conduct an ablation study on PASCAL VOC with a 40% noise ratio using RetinaNet to evaluate the contribution of each metric discussed in Section 4.1. Table 4 summarizes the results under different configurations: (1) without any task-specific metric, (2) using only the class-agnostic (CA) metric for localization, (3) utilizing only the ground-truth box allocation (GTBA) metric for classification, and (4) leveraging both CA and GTBA

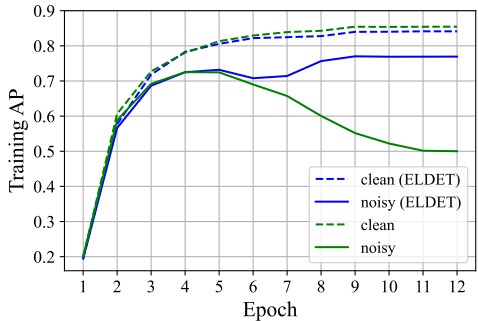

Figure 5: Performance of RetinaNet (green) compared to the setting with ELDET (blue) on the PASCAL VOC training samples. Both models are trained on datasets with 20% combined noisy labels yet evaluated on the original clean annotations. Solid lines represent evaluation on the samples with noisy labels during training, while dashed lines represent evaluation on clean data.

Table 5: Accuracy (%) of noisy label filtering by comparing predictions on noisy samples with the clean labels, evaluated on PASCAL VOC training set using RetinaNet with 20% noise.

| ELDET | Noise Type | |
|---|---|---|
| | Localization | Categorization |
| - | 73.74 | 22.66 |
| ✓ | **80.98** | **74.71** |

Table 6: Ablation study of exponential moving average (EMA) of student parameters to update the teacher network. The results are evaluated on PASCAL VOC using RetinaNet with 40% noise.

| EMA | Noise Type | | |
|---|---|---|---|
| | Localization | Categorization | Combination |
| - | 73.73 | 66.90 | 66.03 |
| ✓ | **74.53** | **73.67** | **68.82** |

metrics. Our final setting with both techniques achieves the best performance, which indicates that task-specific monitoring effectively contributes to robust learning under noisy conditions. The CA manner mitigates the impact of categorization noise, while the GTBA evaluation safeguards detectors against localization noise. The proposed combination of metrics enable ELDET to successfully detect the transition from the early-learning phase to memorization for each task respectively.

**Robustness to Noisy Annotations.** Figure 5 illustrates the performance comparison between RetinaNet and our setting with ELDET when trained on noisy PASCAL VOC but evaluated on the clean labels after each epoch. For example, the solid blue line denotes the *evaluation-with-clean* performance of ELDET on *training-with-noise* samples with 20% noise ratio. This experiment assesses the ability to resist overfitting to noisy annotations and generalize to clean labels. RetinaNet without ELDET shows a significant drop on *evaluation-with-clean* for *training-with-noise* samples (solid green line), which implies the occurrence of memorization to the noisy labels. In contrast, ELDET with high accuracy throughout training demonstrates the robustness in avoiding memorization of noise and its effectiveness in generalizing to clean label even unseen while training. A slight performance drop at the beginning of distillation is observed, which reflects the typical transient instability in early distillation stages caused by conflicting supervision and the delayed stabilization of the EMA teacher.

**Noisy Annotation Filtering for Data Curation.** In addition, we probe the effectiveness of ELDET in validating the quality of annotations on PASCAL VOC using RetinaNet with a noise ratio of 20%. To measure how successfully models identify noisy labels, we compare the model predictions with the original clean ground truth on noisy samples.

In the case of categorization noise, a prediction is considered to be correct if the class prediction is same with the original category. For localization noise, a prediction is valid if IoU compared to the clean box surpasses a threshold of 0.65. The results in Table 5 reveal that detectors with ELDET not only produce accurate predictions on noisy data but also demonstrate a strong capacity to distinguish between clean and noisy annotations. This capability highlights the potential utility of ELDET for identifying and filtering out noisy labels, which suggests practical applications in data curation or data cleaning.

**Effect of EMA on Knowledge Distillation.** We evaluate the impact of the exponential moving average (EMA) on our framework on PASCAL VOC using RetinaNet with a noise ratio of 40%. The teacher network without EMA is not updated after early-learning phase with frozen parameters. Table 6 shows that incorporating EMA significantly improves performance across all noise settings. The improvement in the case of categorization noise shows that the teacher network suffers from limited classification performance in the early-learning phase but that EMA effectively overcomes this limitation. These results underscore the effectiveness of EMA in enhancing the teacher guidance.

Table 7: Performance of ELDET on Deformable DETR under combined noise conditions on PASCAL VOC (AP@50).

| Method | Clean | 20% Noise | 30% Noise | 40% Noise |
|---|---|---|---|---|
| Baseline | 74.27 | 68.60 | 65.39 | 62.52 |
| ELDET | **74.52** | **68.82** | **65.84** | **62.91** |

Table 8: Performance comparison of DINO with query denoising vs. ELDET on PASCAL VOC (40% noise ratio, AP).

| Method | Clean | Loc. | Cat. | Combined |
|---|---|---|---|---|
| Query Denoising | 75.37 | 74.26 | 67.16 | 66.26 |
| ELDET | **76.72** | **75.63** | **67.83** | **68.29** |

Table 9: Hyperparameter sensitivity analysis for our proposed method under different noise conditions. Results are evaluated on the PASCAL VOC dataset using RetinaNet with 40% noise. The table reports performance across various values of $\tau$ and $\gamma$. The best AP scores are highlighted in bold, with the second-best scores underlined.

| $\tau$ | $\gamma$ | Noise Type | | |
|---|---|---|---|---|
| | | Localization | Categorization | Combination |
| 0.1 | 0.9 | 74.53 | **73.67** | 68.82 |
| 0.3 | 0.9 | **76.67** | 70.55 | **73.46** |
| 0.5 | 0.9 | 75.41 | 67.71 | 66.68 |
| 0.1 | 0.7 | 76.11 | 73.07 | 73.39 |
| 0.1 | 0.8 | 75.69 | 73.39 | 71.81 |

**Transformer-based Detectors.** To further demonstrate the generality of ELDET as a plug-and-play module across different architectures, we evaluate it on transformer-based detectors. The results verify ELDET's effectiveness in transformer-specific denoising frameworks and its robustness under challenging noisy conditions. For Deformable DETR [67] on PASCAL VOC with combined noise (as shown in Table 7), ELDET consistently improves performance over the baseline across all noise levels, highlighting its capability to strengthen DETR-based models. Furthermore, to benchmark against native transformer denoising, we replace DINO's query denoising [62] with ELDET and evaluate on PASCAL VOC at 40% noise across all scenarios (Table 8). ELDET surpasses the native method under every noisy condition, with a notable improvement under combined noise (68.29 vs. 66.26 AP). These results validate ELDET's strong adaptability to label noise and confirm its seamless integration into transformer-based detectors such as DINO, extending its applicability beyond convolutional architectures.

**Impact of Hyperparameters** We conduct ablation studies on two key hyperparameters in our ELDET framework: the IoU threshold $\tau$ used in the Ground Truth Box Allocation (GTBA) process, and the deviation threshold $\gamma$ used for early-learning phase detection. Table 9 presents the Average Precision (AP) scores under different settings of $\tau$ and $\gamma$ across localization, categorization, and combined noise types. While certain configurations like $\tau = 0.3$ and $\gamma = 0.9$ achieve the highest AP under localization and combined noise, the first line with $\tau = 0.1$ and $\gamma = 0.9$ provides strong and balanced performance across all noise conditions.

## 6 Conclusion

This paper introduced ELDET, a self-knowledge distillation framework that leverages the early-learning phenomenon to address both localization and categorization noise in object detection. By using early-stage models as teacher networks, ELDET effectively mitigates the memorization of noisy labels, resulting in improved robustness and performance across diverse datasets such as PASCAL VOC, MS COCO and VinDr-CXR. The proposed framework is compatible with various detection architectures, making it practical for real-world applications. Future research could explore extending ELDET to video object detection or integrating it into active learning pipelines for automated data curation.

**Acknowledgments** This work was supported by funding from SAIGE. FC Park was further supported in part by IITP-MSIT grants RS-2021-II212068, 2022-220480, RS-2022-II220480, SNU-AIIS, SNU-IPAI, SNU-IAMD, and the SNU Institute for Engineering Research.

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

# Appendix

## A    Limitations

While our framework demonstrates robust empirical gains under controlled noisy settings, several limitations remain. First, the foundational assumption of our method that localization and categorization exhibit temporally distinct early learning behaviors is based on empirical observations. Although supported by quantitative trends, a theoretical explanation of this phenomenon is beyond the scope of this work and remains an open question. Second, the categorization noise is synthetically generated by uniformly sampling incorrect labels from the set of classes excluding the ground truth. While this is a common strategy in prior work, it does not reflect the structured or semantically biased errors that typically arise in real-world annotation processes. Third, the detection of early-learning phase transitions is performed using curve fitting and gradient slope change, following the heuristic methodology proposed in prior work [33]. However, this procedure is sensitive to the choice of a hyperparameter such as the slope threshold $\gamma$, which can alter the estimated transition point and downstream performance. These limitations underscore the need for more realistic noise modeling, theoretically grounded dynamics analysis, and robust, data-adaptive mechanisms for to identify learning phase transitions.

Table 10: Training resource comparison (training time and GPU memory usage) across various detectors and methods.

| Dataset | Detector | Method | Training time (hours) | Memory Usage (GB) |
|---|---|---|---|---|
| PASCAL VOC (20 classes) | RetinaNet | - | 2.51 | 11,418 |
| | | ORSOD [31] | 3.96 | 12,135 |
| | | ADELE [33] | 4.11 | 11,496 |
| | | ELDET | 4.23 | 17,967 |
| | FCOS | - | 2.13 | 10,507 |
| | | ORSOD [31] | 3.67 | 10,666 |
| | | ADELE [33] | 4.05 | 10,631 |
| | | ELDET | 3.59 | 14,892 |
| MS COCO (80 classes) | RetinaNet | - | 8.03 | 11,418 |
| | | ELDET | 15.43 | 17,967 |

## B    Compute Resources

All experiments were conducted using GPUs with 24GB VRAM (NVIDIA RTX 3090 and 4090). Our framework maintains a teacher model that is initialized at the end of the localization early learning phase and subsequently updated via exponential moving average (EMA) of the student model's parameters. Unlike co-teaching methods that require simultaneous gradient updates to two networks, our approach avoids full dual-model training. Instead, it only requires forward passes through the frozen teacher resulting in moderate memory and compute overhead roughly equivalent to running a single training model alongside a lightweight inference model.

To quantitatively assess computational efficiency, we measured both total training time and GPU memory usage across different configurations. One computational consideration in our setup is the monitoring of early-learning dynamics. To detect phase transitions in learning, we compute validation metrics on the entire training set at every epoch. This process, while crucial to identify transition points accurately, incurs additional time cost especially for large-scale datasets such as MS COCO [30]. This behavior is primarily due to the need for per-epoch validation over the entire training set to detect the early-learning transition point—a step required by all these methods rather than being specific to ELDET.

As shown in Table 10, both ADELE and ORSOD exhibit training time increases comparable to ELDET, primarily due to this per-epoch validation process shared across methods. Although ELDET shows a slightly larger memory footprint, this is mainly due to the additional teacher network used for knowledge distillation. Since the teacher model remains frozen and participates only in forward

passes without backpropagation, the actual computational overhead remains modest. Overall, the additional cost of ELDET is comparable to that of other robust-learning frameworks and is acceptable given the consistent performance improvements observed across datasets.

## C  Detailed Experimental Settings

### C.1  Implementation Details

Our proposed method is implemented using the MMDetection framework [9] built on PyTorch [42]. All input images are resized to $512 \times 512$ for consistency. Training is conducted using Stochastic Gradient Descent (SGD) with a momentum of 0.9 and a weight decay of $10^{-4}$. The learning rate follows a step schedule, decreasing by a factor of 10 at predefined epochs, except for MS COCO [30] where only a linear scheduler is used. For PASCAL VOC [11] and MS COCO, the learning rate is set to 0.01, and the training spans 12 epochs with a batch size of 32. In contrast, for VinDr-CXR [40], the learning rate is set to 0.005, and the training spans 20 epochs with a batch size of 16. We exclude the "No finding" class in VinDr-CXR data for fair comparison of noisy training scenario. All detectors are initialized with a ImageNet [47] pre-trained ResNet-50 [17], and trained on NVIDIA GPUs.

### C.2  ELDET Hyperparameter Details

For the ground-truth box allocation (GTBA), we set the IoU threshold $\tau = 0.1$, replacing predicted box coordinates with ground-truth locations when the IoU exceeds $\tau$. We consider the model to begin memorizing noisy labels when the relative change in the derivative of the performance metric exceeds the criterion with $\gamma = 0.9$. The exponential moving average (EMA) momentum $\alpha$ is set to 0.999 for the overall model and adjusted to $\alpha_{\mathrm{cls}} = 0.1$ for the classification head during the period after localization memorization and before memorizing categorization noise. Other hyperparameters are same as the original setting (*e.g.,* the loss weight $\lambda$ of MMDetection[†].

### C.3  Baselines

ORSOD [31] tackles categorization noise by adopting a dynamic decay mechanism to progressively down-weight the top-$k$ samples with the highest classification loss. The dynamic loss decay function is defined as

$$\mathcal{L}_{\mathrm{DLD}} = \begin{cases} \mathcal{L}_{\mathrm{cls}}(X), & \text{if } t_i < t_{\mathrm{el}}, \\ \alpha \cdot \mathcal{L}_{\mathrm{cls}}(X_k) + \mathcal{L}_{\mathrm{cls}}(X_r), & \text{if } t_i \geq t_{\mathrm{el}}, \end{cases} \tag{8}$$

where $\mathcal{L}_{\mathrm{cls}}$ is the classification loss, $X_k$ and $X_r$ represent the top-$k$ and remaining samples, respectively, $t_i$ denotes the current training epoch, and $t_{\mathrm{el}}$ is the early-learning termination epoch. The decay factor $\alpha$ is defined as:

$$\alpha = \exp\left(-\frac{c}{t_i - t_{\mathrm{el}}}\right), \tag{9}$$

where $c$ is a constant controlling the rate of decay (set to 10 in our experiments). This adaptive mechanism ensures that high-loss samples have reduced impact in later training epochs, which promotes more stable and noise-resilient learning. However, a limitation of ORSOD is that it only suppresses the classification loss without explicitly addressing localization noise in the annotations, which may limit its effectiveness in handling noisy box-level annotations.

ADELE [33] was originally developed for semantic segmentation tasks with noisy annotations, leveraging the observation that early-learning concludes at different times for each class. By updating the labels of pixels where the model's prediction score exceeds a certain threshold (*e.g.,* 0.8) at the class-specific early-learning endpoints, ADELE effectively refines noisy annotations, enabling robust segmentation performance even in the presence of noise. To adapt ADELE for object detection, we modified the approach to account for the inherent differences between segmentation and detection tasks. Instead of utilizing class-specific early-learning endpoints, we defined a unified early-learning endpoint across all classes. At this point, model predictions are used to refine annotations by replacing

---

[†]`https://github.com/open-mmlab/mmdetection`

Table 11: Evaluation on the compatibility with various knowldege distillation techniques. The results are evaluated on PASCAL VOC using RetinaNet with 40% noise. The best AP scores are highlighted in bold, with the second-best scores underlined.

| KD | Noise Type | | |
| --- | --- | --- | --- |
| | Localization | Categorization | Combination |
| - | 70.27 | 68.07 | 65.63 |
| CrossKD [50] | **74.53** | **73.67** | **68.82** |
| FGD [59] | 73.11 | 67.85 | 66.61 |
| OFD [18] | 73.36 | 67.50 | 66.03 |

noisy labels with more reliable predictions that meet strict criteria: (1) a prediction score of at least 0.5, and (2) an Intersection-over-Union (IoU) exceeding 0.5 with the corresponding ground-truth bounding box. For such cases, both the coordinates and the class label of the original ground truth are updated to match the model's prediction.

## C.4 Knowledge Distillation Loss Functions

We adopt the knowledge distillation loss functions used in CrossKD [50] to guide the student models in mimicking the un-memorized knowledge of teacher models. For RetinaNet [46], we use the Quality Focal Loss [26] for classification and the Generalized IoU Loss [45] for localization. In the case of FCOS [49], the classification loss is implemented with Focal Loss [46], while the localization loss employs IoU Loss. For Faster R-CNN [44], the classification loss is based on KL Divergence, and the localization loss uses L1 Loss. Lastly, for GFL [26], the classification loss is also Quality Focal Loss, but the localization loss relies on KD Divergence Loss. These loss functions ensure effective knowledge transfer by aligning the outputs of the student models with those of early-phase teacher models.

## D Additional Experimental Results

### D.1 Compatible with Different Distillation Techniques

To demonstrate the flexibility of our ELDET framework, we investigate its compatibility with various knowledge distillation techniques beyond CrossKD [50]. Specifically, we integrate FGD [59] and OFD [18] into our framework and evaluate their performance under different types of noise.

As shown in Table 11, integrating FGD and OFD into our framework yields improvements over the baseline without distillation under localization and the combined noise. These improvements indicate that our ELDET framework is compatible with different KD techniques and can benefit from them. However, CrossKD consistently outperforms the other distillation methods across all noise types. These results suggest that while our framework can effectively incorporate various KD methods, CrossKD provides the most substantial improvements in our experiments. This superiority may be attributed to CrossKD's ability to facilitate task-oriented knowledge transfer without only focusing on transferring fine-grained feature embeddings from the teacher. Anagnostidis *et al.* [1] found that neural networks are tolerant to label noise except in the last layer, which indicates the vulnerability of the later layers of detectors to noisy annotations. In other words, direct distillation from the classification head of the teacher to that of the student using CrossKD can mitigate the memorization of noisy labels unlike other approaches.

### D.2 EMA Decay Rates

We analyze the impact of the momentum $\alpha$, $\alpha_{\mathrm{cls}}$ and the exponential moving average (EMA) update cycle of the student parameters for the update of the teacher network. Table 12 shows that a small momentum of $\alpha = 0.9$ reduces performance, suggesting that a strong momentum is crucial to maintaining the stability of the teacher network. Similarly, setting $\alpha_{\mathrm{cls}} = 0.999$ or $\alpha_{\mathrm{cls}} = 1.0$ (*i.e.*, updating the classification head slowly before early-learning terminates for classification task) results

Table 12: Evaluation of the EMA (Exponential Moving Average) strategy with varying parameters $\alpha$, $\alpha_{\text{cls}}$, and interval settings. Results are reported as AP@50 on the PASCAL VOC dataset using RetinaNet with 40% combined noise. The table highlights the performance impact of different EMA configurations. The best AP scores are highlighted in bold, with the second-best scores underlined.

| $\alpha$ | $\alpha_{\text{cls}}$ | Interval | AP@50 |
|---|---|---|---|
| 0.999 | 0.1 | 1 | **68.82** |
| 0.999 | 0.1 | 3 | 68.60 |
| 0.999 | 0.1 | 5 | 68.52 |
| 0.9 | 0.1 | 1 | 65.54 |
| 0.999 | 0.999 | 1 | 66.88 |
| 1.0 | 1.0 | 1 | 66.03 |

Table 13: Detection performance comparison under various noise levels on Oxford Pets using RetinaNet. Best AP scores are highlighted in bold.

| ELDET | Noise Level | Localization Noise | Categorization Noise | Combined Noise |
|---|---|---|---|---|
| - | 30% | 89.30 | 79.40 | 84.30 |
| ✓ | 30% | **89.90** | **83.60** | **88.50** |
| - | 50% | 81.00 | 76.80 | 66.50 |
| ✓ | 50% | **85.90** | **78.60** | **79.50** |
| - | 70% | 79.10 | 71.40 | 64.00 |
| ✓ | 70% | **83.10** | **80.10** | **85.90** |

in lower AP. This confirms that using a smaller decay rate $\alpha_{\text{cls}} = 0.1$ for the classification head is important to allow it to adapt more quickly, preventing the teacher from lagging behind the student's learning on classification tasks.

### D.3 Qualitative Examples on VinDr-CXR

Figure 6 presents a qualitative comparison of the detection results of the baseline FCOS [49] and our proposed ELDET method on the VinDr-CXR [40] training set. This comparison underscores the inherent challenges associated with localization and categorization anomalies in medical images. Despite the presence of noisy labels, the detector utilizing ELDET demonstrates significantly better alignment with the ground-truth annotations compared to the baseline FCOS. It highlights the effectiveness of our method in mitigating the adverse effects of both localization and categorization noise. Furthermore, this result emphasizes the robustness of ELDET in diverse domains, demonstrating its applicability not only to real-world images but also to the challenging domain of medical imaging.

### D.4 Results on Smaller Datasets

Although we have already evaluated our method on the relatively small dataset, VinDr-CXR [40], which contains more than ten thousand samples, we further investigated whether the proposed approach remains effective on smaller datasets. We conducted additional experiments on the Oxford Pets [41], which consists of 27 classes with approximately 200 images for each class. As shown in table 13, applying ELDET led to consistently higher performance compared to the baseline without ELDET.

### D.5 Distinctive Early Learning Termination.

We further investigate when the model begins to memorize noisy annotations for localization and classification tasks respectively. As reported in Table 14, models tends to memorize localization noise significantly earlier compared to categorization noise on PASCAL VOC and COCO with various detectors. This observation outlines the necessity of our task-specific guidance mechanism which indicates the appropriate moment to initiate teacher-student distillation for each task.

Table 14: Termination epochs of the localization ($t_\text{loc}$) and classification ($t_\text{cls}$) early-learning phases, and their difference.

| Dataset | Detector | Noise Level | $t_\text{loc}$ | $t_\text{cls}$ | Difference |
|---------|----------|-------------|------|------|------------|
| | | 20% | 3 | 7 | +4 |
| | RetinaNet | 30% | 4 | 4 | +0 |
| | | 40% | 4 | 11 | +7 |
| | | 20% | 3 | 9 | +6 |
| | FCOS | 30% | 4 | 10 | +6 |
| | | 40% | 3 | 11 | +8 |
| PASCAL VOC | | 20% | 7 | 8 | +1 |
| (20 classes) | Faster R-CNN | 30% | 4 | 11 | +7 |
| | | 40% | 3 | 4 | +1 |
| | | 20% | 3 | 8 | +5 |
| | GFL | 30% | 3 | 9 | +6 |
| | | 40% | 6 | 12 | +6 |
| | Average | - | 3.92 | 8.67 | +4.75 |
| | | 20% | 8 | 8 | +0 |
| | RetinaNet | 30% | 4 | 6 | +2 |
| | | 40% | 5 | 9 | +4 |
| | | 20% | 4 | 8 | +4 |
| | FCOS | 30% | 4 | 8 | +4 |
| | | 40% | 3 | 8 | +4 |
| MS COCO | | 20% | 6 | 11 | +5 |
| (80 classes) | Faster R-CNN | 30% | 5 | 11 | +6 |
| | | 40% | 6 | 12 | +6 |
| | | 20% | 7 | 12 | +5 |
| | GFL | 30% | 8 | 8 | +0 |
| | | 40% | 4 | 12 | +8 |
| | Average | - | 5.42 | 9.42 | +4 |

Ground truth    Baseline    ELDET (ours)

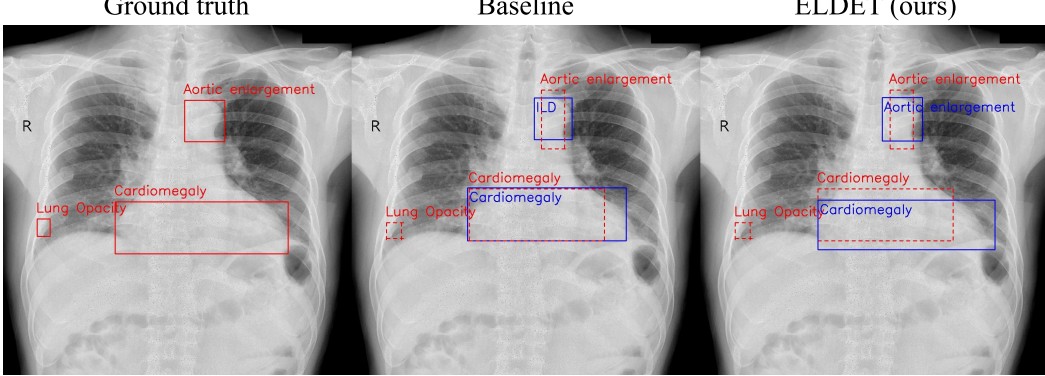

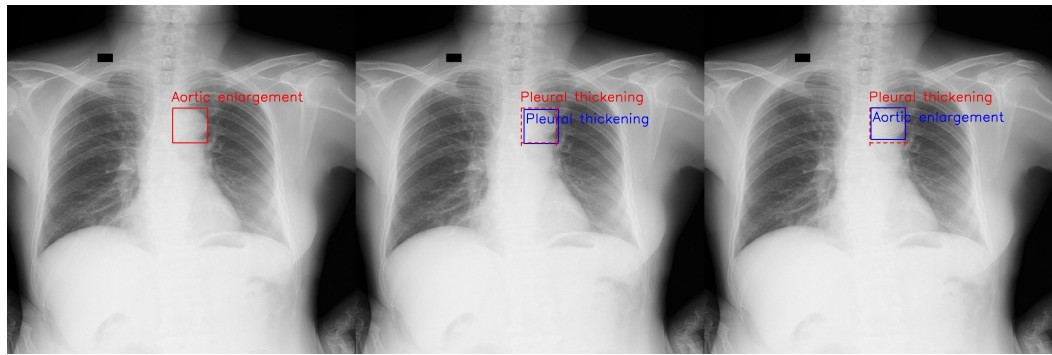

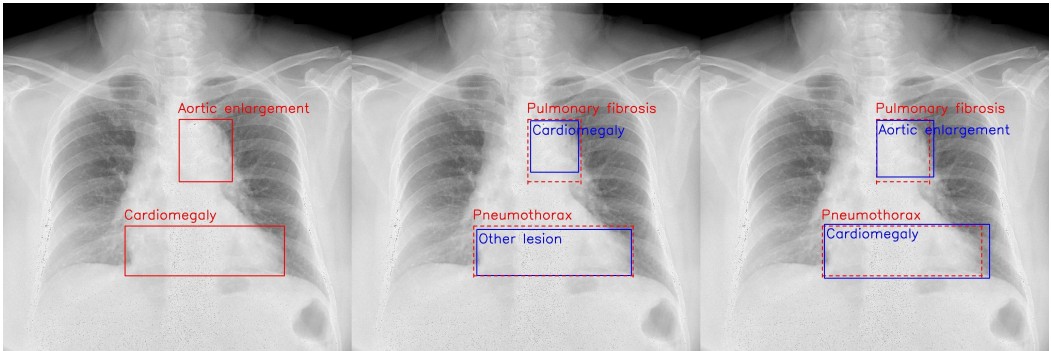

Figure 6: Qualitative Analysis of FCOS on the VinDr-CXR training set. Blue boxes denote model predictions, while red boxes with bold outline represent ground truth annotations. The left panel illustrates the original clean annotations, the middle panel displays predictions from the baseline FCOS, and the right panel shows model outputs with our proposed ELDET. Dotted red boxes in the middle and right panels highlight noisy labels encountered during training. Our proposed ELDET method demonstrates superior capability in mitigating the effects of both localization and categorization noise.

