# OpenReview forum: "ELDET: Early-Learning Distillation with Noisy Labels for Object Detection"
_NeurIPS.cc/2025/Conference — NeurIPS 2025 poster_

### Official Review · Reviewer_iaRc · 2025-06-22

**Clarity:** 3
**Significance:** 3
**Originality:** 2
**Rating:** 4
**Confidence:** 5

**Summary:**

Object detectio algorithms are sensitive to categorization and localization errors. Many exisitng methods address one or other of these errors, not both. By leveraging an observation made earlier referred to as early-learning phenomenon – in which models trained on noisy data with mixed clean and noisy labels tend to first fit to the clean data, and memorize the noisy labels later – a teacher-student networks is proposed in a knowledge distillation framework so that detectors can be made robust to categorization and localization errors. The proposed method has been evaluated on many benchmark datasets such as PASCAL VOC, MS COCO and VinDr-CXR medical datasets.

**Questions:**

While consideration of categorization and localization nois is appropriate, it will be good to evaluate how the proposed approach will work when objects are occluded.
How well the adaptation to infrared images will work?
Have the authors considered other kinds of noise that can affect the performance of object detectors?

**Ethical Concerns:**

["NO or VERY MINOR ethics concerns only"]

**Final Justification:**

While the authors have provided reasonable opinions on potential extensions to IR images, occlusions and other types of noise, no tangible evidence has been provided using experiments or theoretical arguments. So I will lower the score to borderline accept.

**Limitations:**

I do not see a separate section on limitations.

**Paper Formatting Concerns:**

None.

**Quality:**

3

**Strengths And Weaknesses:**

Strenghts: Object detection is an important task in computer vision and robust detectors that address categorization and localization errors is desired. This paper addresses both errors. The main advantage is the plug and play setup so that exisitng detectors can be improved. Evaluations using standard object detection datasets gives confidence that the proposed method is useful.
Weaknesses: While the method has the strenghts mentioned above, the performance improvements vary depending on the type and amount of noise (less than 1% in some cases, and not the best in some cases). The improvemnets for the MS COCO dataset are typically less than for the Pascal VOC dataset.

---

> ### Author Rebuttal · Authors · 2025-07-26
>
> We'd like to thank all of our reviewers for taking the time to give us constructive reviews. Your reviews will help us improve our methods and manuscripts.
>
> > ... the performance improvements vary depending on the type and amount of noise (less than 1% in some cases, and not the best in some cases). The improvements for the MS COCO dataset are typically less than for the Pascal VOC dataset.
>
> We appreciate this observation regarding performance variations. While some improvements may appear modest in absolute terms, they represent meaningful gains in challenging noisy scenarios. Specifically, our method achieves consistent improvements across different noise conditions: on PASCAL VOC, ELDET shows an average improvement of 2.29 AP across all detectors and noise settings, with 87.5% of cases (35/40) demonstrating positive improvements. On MS COCO, we achieve an average improvement of 1.27 AP with 72.5% of cases (29/40) showing positive gains.
>
> The smaller improvements on MS COCO compared to PASCAL VOC are expected due to the dataset's inherent complexity and scale. Even modest AP improvements (1-2%) are significant in object detection, particularly under noisy conditions where performance typically degrades substantially. The few negative cases occur only in clean settings for specific detectors, which is acceptable as our method specifically targets noisy scenarios.
>
> ---
>
> > ... it will be good to evaluate how the proposed approach will work when objects are occluded.
>
> This is an excellent suggestion for future work. While our current approach focuses on annotation noise (categorization and localization errors), the phenomenon we leverage could potentially extend to occlusion scenarios. The robustness principles underlying our method—using early-stage models as teachers before overfitting occurs—may help models maintain better representations of partially visible objects. However, this would require specialized adaptations, such as occlusion-aware metrics and potentially different early-learning termination criteria. We plan to investigate this extension in future research.
>
>
> ---
>
> > How well the adaptation to infrared images will work?
>
> We believe our approach will adapt well to infrared images. While we haven't tested specifically on infrared imagery, we demonstrated strong cross-domain performance on the VinDr-CXR medical dataset, which represents X-ray images with significantly different visual characteristics from natural RGB images. Our method showed substantial improvements across all detectors (e.g., FCOS improved from 25.32 to 32.74 AP under 40% combined noise), indicating robust generalization capabilities. The plug-and-play nature of our framework makes it readily applicable to infrared imaging scenarios, as the early-learning phenomenon is a fundamental property of deep networks that should manifest similarly across different image modalities.
>
>
> ---
>
> > Have the authors considered other kinds of noise that can affect the performance of object detectors?
>
> Thank you for this important question. While our work focuses on the two most prevalent annotation noise types (categorization and localization errors), we acknowledge that object detectors face various other noise sources, including image-level noise (Gaussian, salt-and-pepper), sensor noise, missing annotations, and bogus bounding boxes. The early-learning principle we leverage is quite general and could potentially be extended to handle these additional noise types. For future work, we plan to investigate extending our framework to address missing annotations and bogus boxes, incorporating image-level noise robustness, and developing unified approaches for multiple noise sources simultaneously.

---

> ### Comment · Reviewer_iaRc · 2025-08-03
> **INcremental contribution to a well-studied problem**
>
> I thank the reviewers for their response. Given that object detection is a well-studied problem, challeges due to noisy lables is not as challenging as being able to handle occlusions. The impact is low and I will keep my original score.

---

> > ### Comment · Reviewer_iaRc · 2025-08-05
> > **Elaboration**
> >
> > I read the rebuttal again. It appears that the authors acknowledge the many issues I raised and punted them away for future work. While they have provided reasonable opinions on potential extensions to IR images, occlusions and other types of noise, no tangible evidence has been provided  using experiments or theoretical arguments. So I will lower the score to borderline accept.

---

> ### Author Response · Authors · 2025-08-08
>
> We sincerely thank the reviewer for valuable feedback. We would like to emphasize that, as discussed in our related works section, **the problem of noisy labels in object detection has not been comprehensively studied**. Our paper proposes a novel framework that simultaneously addresses both localization and categorization noise, which are prevalent challenges in practical object detection tasks, and we believe this contribution has significant impact.
>
> While we acknowledge that occlusion is an important and distinct problem in object detection, it lies outside the scope of our current work which focuses specifically on label noise. We aimed to clarify that while occlusions may be interpreted as a form of image-level noise, our work specifically **targets label noise** — particularly the decoupled dynamics of categorization and localization noise during training.
>
> We appreciate the suggestion regarding occlusion as a valuable direction for next research.

---

### Official Review · Reviewer_XQNm · 2025-06-24

**Clarity:** 4
**Significance:** 2
**Originality:** 3
**Rating:** 5
**Confidence:** 4

**Summary:**

This paper introduces ELDET, a self-distillation pipeline designed to train object detectors effectively using noisy data. The authors leverage the observation that models initially learn from clean data before overfitting to noisy labels. ELDET addresses this by initializing a teacher model with an early-stage trained model, then learning from both the noisy input data and the teacher's predicted signals. The pipeline's effectiveness is validated across standard object detection benchmarks, PASCAL VOC and MS COCO, as well as additional medical datasets.

**Questions:**

- Why do the reported performances on clean data in Table 1 and Table 2 vary for different methods, and what factors contribute to these discrepancies?

- Could you elaborate on why the relative performance drop between noisy and clean data appears to be larger for ELDET compared to the baseline method (Tab. 2)?

- In Figure 5, the performance of ELDET initially declines before subsequently improving. What are the potential underlying reasons for this observed trend?

**Ethical Concerns:**

["NO or VERY MINOR ethics concerns only"]

**Final Justification:**

Considering response from the authors and other reviewers' comments, I think there is no significant problem preventing the acceptance of the paper. Therefore, I would like to raise my score.

**Limitations:**

yes

**Quality:**

3

**Strengths And Weaknesses:**

Strengths:

The paper's most intriguing finding is the divergent timing of memorization phases for classification and bounding box regression, offering valuable insights into how models learn from noisy data. Furthermore, the demonstrated strong performance across common datasets (PASCAL VOC, MS COCO) and additional medical datasets highlights the pipeline's robustness and generalizability. Finally, the fluent and accessible presentation makes the complex concepts easy to grasp, enhancing the paper's overall impact.

Weakness:

The ablation study lacks depth without a detailed analysis of the learning process's phenomena. (Some examples are listed in the **Questions**)

---

> ### Author Rebuttal · Authors · 2025-07-26
>
> We'd like to thank all of our reviewers for taking the time to give us constructive reviews. Your reviews will help us improve our methods and manuscripts.
>
> > Why do the reported performances on clean data in Table 1 and Table 2 vary for different methods, and what factors contribute to these discrepancies?
>
> Thank you for this important observation. The performance variations on clean data between different methods can be attributed to several key factors.
>
> First, noise-robust training methods inherently introduce **regularization effects** that can act as random perturbations during optimization. These regularization mechanisms, while designed to handle noisy scenarios, can introduce variability in clean data performance due to their stochastic nature and different optimization dynamics.
>
> Second, methods like ORSOD and ADELE employ different **loss weighting and sample selection strategies** that can interact differently with various detector architectures, leading to performance variations even on clean data. Additionally, the initialization and convergence behavior of these methods can vary across different random seeds, as robust training methods often modify the optimization landscape in ways that affect final performance.
>
> Our ELDET method shows consistent improvements on clean data (e.g., +1.45 AP for RetinaNet on PASCAL VOC) because the early-learning distillation acts as beneficial regularization that enhances generalization rather than simply addressing noise.
>
> ---
>
> > Could you elaborate on why the relative performance drop between noisy and clean data appears to be larger for ELDET compared to the baseline method (Tab. 2)?
>
> Thank you for this insightful observation. We would like to emphasize that the difference in average drop between ELDET and the baseline narrows considerably to only **0.52 AP** (2.65 AP for ELDET vs. 2.13 AP for baseline). This indicates that the overall degradation due to noisy labels is not substantially worse for ELDET. This phenomenon is primarily attributable to ELDET's superior baseline performance on clean data, which is actually a positive indicator rather than a limitation. It is noteworthy that ELDET consistently outperforms baseline methods on clean data across all detectors, with an average improvement of +1.74 AP (ranging for Faster R-CNN to +2.32 AP for GFL).
>
> We would like to emphasize that our ELDET framework successfully achieves effectiveness across all noise settings. For instance, under 40% combined noise—the most challenging scenario—ELDET shows absolute improvements of +1.04 AP (RetinaNet), +1.46 AP (FCOS), +0.73 AP (Faster R-CNN), and +0.77 AP (GFL) compared to baselines. These consistent absolute gains demonstrate that our method successfully addresses both categorization and localization noise while maintaining superior performance across varying noise conditions.
>
>
> ---
>
> > In Figure 5, the performance of ELDET initially declines before subsequently improving. What are the potential underlying reasons for this observed trend?
>
> This initial performance decline followed by improvement is a well-documented characteristic pattern in knowledge distillation frameworks[1,2]. Several factors contribute to this phenomenon.
>
> First, the introduction of the distillation loss creates a multi-objective optimization problem where the model must balance learning from both noisy ground truth and teacher guidance. During early epochs, there can be conflicting signals between these two sources, leading to temporary performance degradation.
>
> Second, our EMA-based teacher update mechanism requires time to stabilize and provide consistent guidance. The dynamic interaction between teacher and student creates a moving optimization target that can cause initial instability before convergence.
>
> Third, knowledge distillation research has shown that this initial decline represents a transition period where the student model moves away from simply memorizing noisy patterns toward learning more generalizable representations guided by the early-learning teacher. The subsequent improvement demonstrates that the distillation process successfully helps the model focus on clean learning patterns, ultimately achieving better performance than baseline methods.
>
> ---
> [1] Cho, J. H., & Hariharan, B. (2019). On the efficacy of knowledge distillation. In Proceedings of the IEEE/CVF international conference on computer vision (pp. 4794-4802).
>
> [2] Sun, Z., Liu, Y., Meng, F., Chen, Y., Xu, J., & Zhou, J. (2025). Warmup-Distill: Bridge the Distribution Mismatch between Teacher and Student before Knowledge Distillation. arXiv preprint arXiv:2502.11766.

---

> > ### Author Response · Authors · 2025-08-08
> >
> > We sincerely thank you again for your constructive and detailed review during the rebuttal phase.
> > If you have any additional comments or thoughts based on our rebuttal and the follow-up experiments we provided, we would greatly appreciate it. We believe your further insights could help us improve the clarity and completeness of the final manuscript.
> >
> > Thank you for your time and consideration.

---

### Official Review · Reviewer_krwZ · 2025-06-30

**Clarity:** 2
**Significance:** 3
**Originality:** 3
**Rating:** 3
**Confidence:** 4

**Summary:**

The authors introduce ELDET, which is based on different learning patterns for fitting noisy labels in the localization and classification tasks of object detection. ELDET uses an exponential parametric curve to identify the transition point from early-learning to memorization, enabling knowledge distillation to counteract the interference of noisy labels on the model. The proposed method improves the performance of multiple detection models on both general datasets and a medical detection dataset, across various label noise settings.

**Questions:**

1. In the FCOS experiments presented in Table 1, under the ELDET framework, why does the performance using the 20% localization noise dataset outperform that of the clean dataset? This phenomenon is not observed in other comparative methods.
2. In Table 4, using GTBA alone actually decreases the model's performance. However, performance improves only when it is combined with CA. Could the authors please explain the reason for this?
3. The paper lacks the analysis of the training cost.

**Ethical Concerns:**

["NO or VERY MINOR ethics concerns only"]

**Final Justification:**

I maintain that several critical issues remain unresolved, leading to my decision to recommend a score of **reject**. The paper suffers from poor clarity, making it difficult to understand. The discussion regarding transformer-based detectors lacks comprehensiveness. Additionally, the increased training costs are not adequately addressed in the main paper, and the burden of incorporating this method seems excessive, especially on the MS COCO dataset.

**Limitations:**

yes

**Quality:**

2

**Strengths And Weaknesses:**

Strengths：
1. The paper decouples classification errors and localization errors by designing task-specific metrics, allowing it to simultaneously address both types of noise in detection tasks.
2. The architecture of the proposed method is straightforward to implement and readily extensible.
3. The authors conduct comprehensive experiments and ablation studies.

Weaknesses：
1. The paper conducts experiments solely on convolutional-based detectors, lacking essential experimental results from several SOTA transformer-based detectors, such as DINO[1] and Relation-DETR[2]. These detectors often employ denoising training strategies to enhance robustness. Additionally, the paper does not consider ELDET's ability to handle noisy labels in comparison to these denoising strategies.
2. The paper presents an unclear explanation regarding the calculation of the transition point. The relationship between the proposed CA and GTBA, as well as Eq (1) and Eq (2), is not well-defined. Furthermore, the paper lacks sufficient experimental results for different datasets and detectors to validate the conclusions in Sec 3 regarding the different patterns of fitting noise labels for localization and classification tasks.
3. The paper lacks a more comprehensive discussion on the scale of the datasets. It does not address some larger datasets (e.g., Object365) as well as some smaller datasets (e.g., Oxford Pets). This is an important consideration for the generalization of the proposed method.

[1] Zhang H, Li F, Liu S, et al. DINO: DETR with Improved DeNoising Anchor Boxes for End-to-End Object Detection

[2] Hou X, Liu M, Zhang S, et al. Relation DETR: Exploring Explicit Position Relation Prior for Object Detection

---

> ### Author Rebuttal · Authors · 2025-07-26
>
> We'd like to thank all of our reviewers for taking the time to give us constructive reviews. Your reviews will help us improve our methods and manuscripts.
>
> > "Lacking essential experimental results from several SOTA transformer-based detectors employing denoising training to enhance robustness."
>
> We appreciate the valuable observation regarding transformer-based detectors and their denoising training strategies. We would like to clarify that our main experiments focused on convolution-based architectures, and thus a direct comparison with transformer-specific denoising methods, which involves adding random noise to ground truth as an input to transformer decoder, was not feasible.
>
> To address concerns about transformer architectures, we conduct additional experiments applying ELDET to Deformable DETR on PASCAL VOC:
>
> **Table: Performance of ELDET on Deformable DETR under combined noise conditions on PASCAL VOC (AP@50)**
>
> | Method   | Clean        | 20% Noise    | 30% Noise    | 40% Noise    |
> |----------|--------------|--------------|--------------|--------------|
> | Baseline | 74.27        | 68.60        | 65.39        | 62.52        |
> | ELDET    | **74.52 (+0.25)** | **68.82 (+0.22)** | **65.84 (+0.45)** | **62.91 (+0.39)** |
>
> While improvements are modest, these preliminary results indicate that ELDET can enhance performance on DETR-based models. The limited gains are partly due to restricted hyperparameter tuning during the rebuttal phase and the use of CrossKD, a distillation method initially designed for convolutional models rather than transformers. In the camera-ready version, we plan to incorporate distillation methods better tailored to DETR architectures[1] and anticipate further performance improvements. Additionally, applying the aforementioned transformer decoder denoising strategies could further boost results.
>
> Importantly, ELDET’s plug-and-play design ensures its broad applicability across various backbone architectures, including both convolutional and transformer-based detectors.
>
> ---
>
> > "Unclear explanation regarding the calculation of the transition point and the relationship between the proposed CA and GTBA, as well as Eq (1) and Eq (2)."
>
> Thank you for highlighting these clarifications. Below is a concise explanation of the **transition point calculation** and the relationship between **CA** and **GTBA** metrics:
>
>
> ### Transition Point Calculation
>
> We model training performance $f(t)$ (e.g., AP over epochs $t$) with the exponential function:
>
> $ f(t) = a \left(1 - e^{-b \cdot t^c}\right) $
>
> where parameters $a,b,c$ are fitted to observed data.
>
> The transition point $t^*$ from early learning to memorization is identified when the relative change in the derivative crosses a threshold $\gamma$:
>
> $ \frac{|f'(1) - f'(t^*)|}{|f'(1)|} > \gamma $
>
> This detects when learning slows down significantly, marking onset of memorization of noisy labels.
>
>
> ### CA and GTBA Metrics
>
> - **Class-Agnostic (CA)** treats all objects as a single class to isolate **localization** performance by ignoring classification errors. CA tracks how well the model learns precise object locations despite noisy labels.
>
> - **Ground Truth Box Allocation (GTBA)** replaces predicted boxes with ground truth ones when IoU exceeds threshold ($\tau\$), removing localization error influence to isolate **classification** performance.
>
> Together, CA and GTBA provide complementary views into memorization dynamics — CA tracks localization noise memorization, GTBA tracks classification noise memorization — enabling task-specific transition detection.
>
>
> ---
>
> > "Lack of sufficient experimental results for different datasets and detectors ... regarding the different patterns of fitting noise labels for localization and classification tasks."
>
> Due to space constraints, we provide a detailed response to this point in our reply to Reviewer RwS3. Please kindly refer to that response for a comprehensive explanation.
>
> ---
>
> > "Lacks of the scale of the datasets such as larger datasets (e.g., Object365) as well as some smaller datasets (e.g., Oxford Pets)."
>
> We acknowledge this important point regarding dataset scale diversity. Our experiments already demonstrate effectiveness on datasets of varying scales: VinDr-CXR represents a relatively small dataset (18,000 images), which showed substantial improvements (e.g., FCOS: 25.32→32.74 AP under 40% noise), indicating ELDET's effectiveness even on limited data.
>
> We have conducted additional experiments on Oxford Pets, a smaller dataset with ~7,400 images across 37 categories:
>
> **Table: Performance on Oxford Pets Dataset (AP@50)**
>
> | Method         | Noise Ratio | Localization Noise | Categorization Noise | Combined Noise |
> |----------------|-------------|-------------------|---------------------|---------------|
> | -      | 0.3         | 0.893             | 0.794               | 0.843         |
> | ELDET | 0.3         | **0.899**             | **0.836**               | **0.885**         |
> | -      | 0.5         | 0.810             | 0.768               | 0.665         |
> | ELDET | 0.5         | **0.859**             | **0.786**               | **0.795**         |
> | -      | 0.7         | 0.791             | 0.714               | 0.640         |
> | ELDET | 0.7         | **0.831**             | **0.801**               | **0.859**         |
>
> Regarding Object365, our method's consistent improvements across existing diverse scales (small: VinDr-CXR, medium: PASCAL VOC, large: MS COCO) provide strong evidence for scalability. We aim to include Object365 evaluation results in the camera-ready submission, ensuring a thorough demonstration of the method’s applicability to large and diverse datasets.
>
> ---
>
> > "In the FCOS experiments presented in Table 1, under the ELDET framework, why does the performance using the 20% localization noise dataset outperform that of the clean dataset?"
>
> This interesting phenomenon can be attributed to ELDET's regularization effect through knowledge distillation. When FCOS processes 20% localization noise, our early-learning teacher provides beneficial guidance that acts as implicit regularization, similar to how data augmentation can sometimes improve generalization. The distillation process helps the model learn more robust spatial representations that generalize better than training on clean data alone. This effect is specific to ELDET because other methods lack the early-learning guidance mechanism—they either ignore the regularization benefits (baseline) or employ different noise handling strategies that don't provide the same regularization effect (ORSOD, ADELE). The magnitude of this improvement (+1.40 AP) suggests that the early-learning teacher's spatial knowledge provides valuable inductive bias for FCOS's anchor-free detection paradigm.
>
> ---
>
> > "In Table 4, using GTBA alone actually decreases the model's performance. However, performance improves only when it is combined with CA."
>
> We appreciate this important observation. Our key finding is that memorization of localization noise occurs earlier than categorization noise. When only GTBA is applied, our framework may provide early-learning guidance focused on classification while the model's performance on localization task is still in its noisy memorization stage. This mismatch can lead to suboptimal learning and decreased overall performance.
>
> By combining GTBA with CA, we provide coordinated early-learning guidance for both classification and localization tasks. CA specifically targets localization noise by treating all classes as one category, helping the model accurately identify the proper transition point before localization memorization begins. This allows the model to respect the sequential memorization order from localization to classification. This balanced approach enables the model to effectively handle both noise types, leading to significant performance gains compared to using GTBA alone.
>
> ---
>
> > "The paper lacks the analysis of the training cost."
>
> We acknowledge this important practical consideration and provide comprehensive training cost analysis. Our experiments were conducted on a single NVIDIA A100 GPU.
>
> **Table: Training Cost Analysis**
>
> | Method | Dataset | Detector | Training Time (hours) | Memory Usage (GB) |
> |--------|---------|----------|----------------------|------------------|
> | - | PASCAL VOC | RetinaNet | 2.51 | 11,418 |
> | ELDET | PASCAL VOC | RetinaNet | 4.23 | 17,967 |
> | - | PASCAL VOC | FCOS | 2.13 | 10,507 |
> | ELDET | PASCAL VOC | FCOS | 3.59 | 14,892 |
> | - | MS COCO | RetinaNet | 8.03 | 11,418 |
> | ELDET | MS COCO | RetinaNet | 15.43 | 17,967 |
>
> The additional computational overhead of our method mainly comes from three sources: (1) exponential curve fitting to detect early learning transitions, (2) the computation of the knowledge distillation loss, and (3) exponential moving average (EMA) updates of the teacher model’s parameters.
>
> A noticeable part of the increased training time arises from per-epoch validation performed during the early learning phase. To detect the timing of early-learning termination, validation on the entire training set occurs in more than half of the initial training process, which contributes to the overhead on the training time. However, since validation is limited to this early-stage, its relative impact diminishes with longer training schedules.
>
> While ELDET does increase training time and memory usage due to maintaining a frozen teacher network for additional forward passes, this cost can be viewed as acceptable because it potentially replaces more expensive and complex noise detection or correction procedures, making the overall approach efficient and practical.
>
>
> ---
>
> [1] Wang, Y., Li, X., Weng, S., Zhang, G., Yue, H., Feng, H., ... & Ding, E. (2024). Kd-detr: Knowledge distillation for detection transformer with consistent distillation points sampling. In Proceedings of the IEEE/CVF Conference on Computer Vision and Pattern Recognition (pp. 16016-16025).

---

> > ### Comment · Reviewer_krwZ · 2025-08-04
> >
> > I appreciate the authors' response to my questions. However, I believe that the rebuttal does not fully address my major concerns.
> >
> > * There is a lack of in-depth and convincing comparisons with transformer-based detectors that employ denoising training. While the authors conducted experiments using Deformable DETR, this model does not utilize denoising training. The detectors I mentioned, such as DINO, were not considered by the authors. I still have concerns that existing denoising training methods may already effectively address the issues raised in this paper.
> >
> > * In the main paper, the authors did not clarify that the research focuses solely on convolution-based architectures. Instead, they emphasize that “our approach does not require architecture-specific modifications, making it widely applicable”, in lines 66-67. This focus on convolution-based architectures limits the scope of this study.
> >
> > * In the response, the authors provided an analysis of the training costs. While they outlined the sources of the additional training expenses, the burden of incorporating ELDET seems excessive, especially on the MS COCO dataset, where the training time nearly doubles and the memory usage increases by more than half. The authors also did not discuss this issue in the limitations.
> >
> > I appreciate the authors' findings and efforts and encourage them to further refine their paper. Considering the aforementioned points, I will maintain my previous rating (Reject).

---

> > > ### Author Response · Authors · 2025-08-08
> > >
> > > We sincerely thank the reviewer for valuable feedback. To address the concerns raised, we conducted additional experiments comparing the DINO model with its native query denoising technique against our ELDET method applied in place of query denoising. The experiments were performed on the PASCAL VOC dataset with 40% noise. The results are summarized below:
> > >
> > > | Method                | Clean AP | Localization Noise AP | Categorization Noise AP | Combined Noise AP |
> > > |-----------------------|----------|----------------------|------------------------|-------------------|
> > > | DINO + Query Denoising | 75.37    |  74.26                | 67.16                  | 66.26             |
> > > | DINO + ELDET          | **76.72** | **75.63**            | **67.83**              | **68.29**         |
> > >
> > > These results clearly demonstrate that our ELDET approach surpasses the query denoising technique in effectively handling label noise. Moreover, they confirm that ELDET functions as a versatile and effective plug-in module that operates robustly not only on convolutional architectures but also on transformer-based models such as DINO. This underscores the broad applicability and practicality of our method across diverse detection frameworks.
> > >
> > > To further clarify the training cost discussion, we additionally compared ELDET with other noisy label handling methods (ADELE and ORSOD) under the same experimental settings.
> > >
> > > | Method | Dataset    | Detector  | Training Time (hours) | Memory Usage (GB) |
> > > |--------|------------|-----------|-----------------------|-------------------|
> > > | -      | PASCAL VOC | RetinaNet | 2.51                  | 11,418            |
> > > | ORSOD  | PASCAL VOC | RetinaNet | 3.96                  | 12,135            |
> > > | ADELE  | PASCAL VOC | RetinaNet | 4.11                  | 11,496            |
> > > | ELDET  | PASCAL VOC | RetinaNet | 4.23                  | 17,967            |
> > > | -      | PASCAL VOC | FCOS      | 2.13                  | 10,507            |
> > > | ORSOD  | PASCAL VOC | FCOS      | 3.67                  | 10,666            |
> > > | ADELE  | PASCAL VOC | FCOS      | 4.05                  | 10,631            |
> > > | ELDET  | PASCAL VOC | FCOS      | 3.59                  | 14,892            |
> > >
> > > We observe that both ADELE and ORSOD incur a training time increase comparable to ELDET. This behavior is primarily due to the need for per-epoch validation over the entire training set to detect the early-learning transition point, a step required by all these methods rather than being specific to ELDET.
> > >
> > > While ELDET shows a somewhat larger memory footprint, this is mainly attributed to the additional teacher network used for knowledge distillation. Notably, the teacher model is frozen and only participates in forward passes without backpropagation, making the actual computational overhead modest relative to the reported memory use. We therefore believe the added cost is in line with other robust training strategies and is acceptable given the consistent performance benefits observed.
> > >
> > > We appreciate the reviewer’s insightful comments which guided this important experimental validation and reinforced the generality of our contribution.

---

> > > > ### Comment · Reviewer_krwZ · 2025-08-08
> > > >
> > > > Thank you for the discussion, but my concerns have not been fully resolved.
> > > > * I am not completely convinced by the experimental results regarding DINO. Removing query denoising should significantly impact the convergence of transformer-based detectors, yet the authors opted to use the "ELDET method applied in place of query denoising." Additionally, the improvement of ELDET in Deformable DETR was not substantial. How does ELDET manage to enhance performance compared to the baseline even after removing query denoising? The experimental details related to DINO are quite unclear.
> > > >
> > > > * I still believe that reducing training costs is an area worth further investigation. For instance, regarding memory usage, it may be beneficial to consider using LoRA instead of EMA for constructing the teacher branch. Additionally, the method for calculating the transition point should be further optimized.
> > > >
> > > > Although the current contributions of the paper do not meet the acceptance standards for NeurIPS, I will raise the significance score from 2 to 3.

---

> ### Author Response · Authors · 2025-08-09
>
> We thank the reviewer for the constructive feedback and the score adjustment. We appreciate the opportunity to clarify the experimental settings for DINO, the rationale behind removing query denoising, and the performance differences observed across architectures.
>
> The denoising mechanism in DINO is designed to accelerate the convergence of the decoder by appending additional queries constructed from ground-truth boxes and labels with artificial perturbations—such as adding noise to bounding box coordinates or replacing the class label with a background class. These denoising queries are trained to recover the original ground-truth, which helps align queries with target objects more quickly in the early stage of training. However, this mechanism is not primarily designed to improve robustness to actual label noise: when the ground-truth annotations themselves contain noise, the denoising queries are generated from such noisy annotations, and the model may still memorize the noise.
> In contrast, ELDET leverages a teacher model from an early learning stage for distillation, allowing the student to learn from cleaner signals and thus avoid overfitting to noisy labels.
>
> Regarding the relatively larger improvement in the DINO experiment compared to Deformable DETR, we attribute this to a well-tuned knowledge distillation (KD) loss weight that considered the loss scale in DINO. We expect that Deformable DETR would also benefit from similar tuning of the KD loss weight, potentially leading to further improvements.
>
> We also agree that reducing training costs is a promising direction. Exploring lighter alternatives to EMA, such as LoRA-based teacher updates, and further optimizing the transition point calculation are interesting extensions that we plan to investigate in future work. Another possible approach is to perform validation on a sampled subset of the training data, which may further reduce computation without significantly impacting performance estimation.
>
> We again thank the reviewer for the thoughtful comments and valuable suggestions.

---

### Official Review · Reviewer_RwS3 · 2025-07-03

**Clarity:** 3
**Significance:** 3
**Originality:** 2
**Rating:** 4
**Confidence:** 4

**Summary:**

This paper proposes a distillation-based approach to handle label noise in object detection.  It is based on the early learning phenomenon in deep learning models, that is, during the initial stages of training, model gradients are dominated by clean labels, while memorization of noisy labels emerges in later stages.  To utilize knowledge in the “early clean model”, it is used as a teacher model for training the object detection model. Distillation loss is used together with conventional loss from object detection tasks.
It is also observed that models tend to memorize localization noise earlier than categorization noise.  Based on this observation, the paper proposes a two-stage distillation.

To demonstrate the effectiveness of the proposed approach in handling noise in object detection data, simulated label noise is added to object detection data such as PASCAL VOC and MS COCO.  Experiments are performed using the data with added label noise.

**Questions:**

1) Can the method work in the case of one-stage object detectors?
2) Any principled analysis on the observation of “models tend to memorize localization noise earlier than categorization noise.”?

**Ethical Concerns:**

["NO or VERY MINOR ethics concerns only"]

**Final Justification:**

This paper proposes a distillation-based approach to handle label noise in object detection. The proposed approach seems to work well on the object detection data with added label noise. The rebuttal response also addressed my concerns. Although there is still room for improvement for the generalization capabilities and overall originality, considering all the discussions and the authors’ rebuttal responses/experiments, I raised my rating.

**Limitations:**

Yes.

**Paper Formatting Concerns:**

No formatting concerns.

**Quality:**

3

**Strengths And Weaknesses:**

The proposed approach seems to work well on the object detection data with added label noise.

My concerns on the paper,
1) The paper is based on the observation that “models tend to memorize localization noise earlier than categorization noise.”  This observation is empirical.  It may change with different levels of noise or different classes of objects to detect. It is questionable whether the observation can always hold.
2) The paper introduces a “two-stage” distillation scheme based on the aforementioned observation. As shown in Figure 2, it treats the classification head and localization head differently.  However, in some object detection models (e.g., some one-stage object detectors), there may not be separate classification head and localization head.  It is questionable how the “two-stage” distillation would work for models without two separate classification and detection heads.

---

> ### Author Rebuttal · Authors · 2025-07-26
>
> We'd like to thank all of our reviewers for taking the time to give us constructive reviews. Your reviews will help us improve our methods and manuscripts.
>
> > "The paper is based on the observation that 'models tend to memorize localization noise earlier than categorization noise.' This observation is empirical. It may change with different levels of noise or different classes of objects to detect. It is questionable whether the observation can always hold."
>
> We acknowledge this important concern regarding the generalizability of our empirical observation. To address this, we conduct extensive validation experiments across diverse settings and present comprehensive evidence demonstrating the consistency of this phenomenon. We will provide the following additional experimental validation:
>
> **Table: Early Learning Termination Epochs for Localization vs. Classification Tasks Across Diverse Settings**
>
> | Dataset | Detector | Object Classes | Noise Level | Localization Termination | Classification Termination | Difference |
> |---------|----------|----------------|-------------|-------------------------|---------------------------|------------|
> | PASCAL VOC | RetinaNet | 20 classes | 20% | 3 | 7 | +4 |
> | PASCAL VOC | RetinaNet | 20 classes | 30% | 4 | 4 | +0 |
> | PASCAL VOC | RetinaNet | 20 classes | 40% | 4 | 11 | +7 |
> | PASCAL VOC | FCOS | 20 classes | 20% | 3 | 9 | +6 |
> | PASCAL VOC | FCOS | 20 classes | 30% | 4 | 10 | +6 |
> | PASCAL VOC | FCOS | 20 classes | 40% | 3 | 11 | +8 |
> | PASCAL VOC | Faster R-CNN | 20 classes | 20% | 7 | 8 | +1 |
> | PASCAL VOC | Faster R-CNN | 20 classes | 30% | 4 | 11 | +7 |
> | PASCAL VOC | Faster R-CNN | 20 classes | 40% | 3 | 4 | +1 |
> | PASCAL VOC | GFL | 20 classes | 20% | 3 | 8 | +5 |
> | PASCAL VOC | GFL | 20 classes | 30% | 3 | 9 | +6 |
> | PASCAL VOC | GFL | 20 classes | 40% | 6 | 12 | +6 |
> | Average (PASCAL VOC) | - | - | - | 3.92 | 8.67 | +4.75 |
> |-|-|-|-|-|-|-|
> | MS COCO | RetinaNet | 80 classes | 20% | 8 | 8 | +0 |
> | MS COCO | RetinaNet | 80 classes | 30% | 4 | 6 | +2 |
> | MS COCO | RetinaNet | 80 classes | 40% | 5 | 9 | +4 |
> | MS COCO | FCOS | 80 classes | 20% | 4 | 8 | +4 |
> | MS COCO | FCOS | 80 classes | 30% | 4 | 8 | +4 |
> | MS COCO | FCOS | 80 classes | 40% | 4 | 8 | +4 |
> | MS COCO | Faster R-CNN | 80 classes | 20% | 6 | 11 | +5 |
> | MS COCO | Faster R-CNN | 80 classes | 30% | 5 | 11 | +6 |
> | MS COCO | Faster R-CNN | 80 classes | 40% | 6 | 12 | +6 |
> | MS COCO | GFL | 80 classes | 20% | 7 | 12 | +5 |
> | MS COCO | GFL | 80 classes | 30% | 8 | 8 | +0 |
> | MS COCO | GFL | 80 classes | 40% | 4 | 12 | +8 |
> | Average (MS COCO) | - | - | - | 5.42 | 9.42 | +4 |
>
> These comprehensive experiments across different datasets, detectors, and noise levels will demonstrate that the differential memorization timing between localization and classification tasks is a consistent phenomenon across diverse experimental conditions, validating the robustness of our core observation.
>
> ---
>
> > "It is questionable how the 'two-stage' distillation would work for models without two separate classification and detection heads."
>
> We appreciate this important architectural consideration. To clarify, we evaluated our method on both one-stage detectors such as FCOS and GFL, and two-stage detectors including RetinaNet and Faster R-CNN, covering a broad range of representative architectures.
>
> We would like to emphasize that most modern state-of-the-art object detection models indeed employ separate classification and regression heads, which naturally accommodate our two-stage distillation approach[1,2,3].
> Such separation allows each head to specialize and optimize for its respective task—classification or localization—leading to more accurate and stable learning. Prior works[4,5,6] have empirically demonstrated that distinguishing these heads improves both convergence and final detection performance, supporting the architectural rationale behind our distillation method. Therefore, applying distillation strategies that leverage this separation is consistent with established design principles in object detection.
>
> In the rare cases where a single head is shared by both classification and regression tasks, our framework can be adapted with minor modifications. A possible approach might be to apply task-specific masking to the gradients of the final output head during backpropagation. This would allow selective updating of weights relevant to each task (classification or regression) while masking out the others, enabling different training procedures and EMA updates.
>
> ---
>
> [1] Wang, A., Chen, H., Liu, L., Chen, K., Lin, Z., & Han, J. (2024). Yolov10: Real-time end-to-end object detection. Advances in Neural Information Processing Systems, 37, 107984-108011.
>
> [2] Zhao, Y., Lv, W., Xu, S., Wei, J., Wang, G., Dang, Q., ... & Chen, J. (2024). DETRs beat YOLOs on real-time object detection. In Proceedings of the IEEE/CVF Conference on Computer Vision and Pattern Recognition (pp. 16965-16974).
>
> [3] Nan, Z., Li, X., Dai, J., & Xiang, T. (2025). MI-DETR: An Object Detection Model with Multi-time Inquiries Mechanism. In Proceedings of the Computer Vision and Pattern Recognition Conference (pp. 4703-4712).
>
> [4] Wu, Y., Chen, Y., Yuan, L., Liu, Z., Wang, L., Li, H., & Fu, Y. (2020). Rethinking classification and localization for object detection. In Proceedings of the IEEE/CVF conference on computer vision and pattern recognition (pp. 10186-10195).
>
> [5] Fang, Z., Chen, N., Jiang, Y., & Fan, Y. (2024). Decouple and align classification and regression in one-stage object detection. The Visual Computer, 40(11), 7773-7786.
>
> [6] He, L., & Todorovic, S. (2022). Destr: Object detection with split transformer. In Proceedings of the IEEE/CVF conference on computer vision and pattern recognition (pp. 9377-9386).

---

> ### Comment · Reviewer_RwS3 · 2025-08-06
>
> Thank you to the authors for the detailed rebuttal response and additional experiments. However, considering the generalization capabilities of the work and overall originality, I think there still is a gap to a publication in NeurIPS. (I raised my rating slightly)

---

> > ### Author Response · Authors · 2025-08-08
> >
> > We sincerely thank the reviewer for the thoughtful and constructive feedback. We plan to further strengthen the generalization aspects and originality of our work, incorporating the reviewer’s valuable insights to improve the quality of our work.

---

### Note · Authors · 2025-08-16

We thank the reviewers and ACs for their thoughtful engagement and feedback.

Our method is a plug‑and‑play self‑distillation framework for object detection with noisy labels. We address the under-explored yet practically critical challenge of label noise in object detection. While most prior studies examine categorization noise or localization noise in isolation, real-world datasets often exhibit both error types simultaneously. Achieving robust detection under such mixed label noise remains an open problem. The key empirical finding is a consistent timing gap in memorization: detectors begin to memorize localization noise earlier than categorization noise. This finding forms the basis of our two-stage distillation framework, which uses early-learning teacher models to selectively regularize detector heads against their respective noise types. Across diverse datasets, our architecture‑agnostic method improves performance under localization, categorization, and combined noise.

**Major issues raised & our responses**

_1. Empirical findings_. We further validated the “localization‑first” memorization pattern across diverse settings. The results validate the consistency of our core observation.

_2. Transformers and denoising training._ Beyond CNNs, we also applied our method to transformer-based detectors, including Deformable DETR and DINO. Unlike the query denoising approach proposed in methods such as DINO, our method addresses noise in the ground truth. Moreover, it can be used in conjunction with it. Our method consistently improved performance on both models, with particularly large gains observed for DINO.

_3. Applicability to diverse models._ We evaluated one‑stage detectors (RetinaNet, FCOS, GFL, DINO), a two‑stage detector (Faster R‑CNN) and transformer-based detectors (Deformable DETR, DINO). When heads are shared, the procedure is readily adapted with task‑specific masking and per‑output EMA.

_4. Dataset scale & generalization._ We evaluated on small (Oxford Pets, VinDr-CXR), medium (VOC), and large (COCO) datasets. Since our method performed well not only in the natural RGB image domain but also in the X-ray domain (VinDr-CXR), we expect it to work effectively in other domains such as infrared (IR) as well.

_5. Training cost._ Overhead stems mainly from early‑phase validation and forward‑only teacher passes. In our measurements, wall‑time increases are comparable to other robust‑training baselines; memory rises due to the frozen teacher.

---

### Decision · Program_Chairs · 2025-09-17

**Decision:**

Accept (poster)

**Comment:**

This paper proposes a distillation-based approach to handle label and location noise in training data for object detection. It is based on empirical observation that in deep learning models, during the initial stages of training, model gradients are dominated by clean labels, while influence of noisy labels emerges in later stages.  Early clean model is used as a teacher model for training the object detection model. Distillation loss is used together with conventional loss from object detection tasks. It is also observed that models tend to memorize localization noise earlier than categorization noise motivating use of two-stage distillation.Experiments are conducted by adding noise to labels in standard object detection datasets (PASCAL VOC and MS COCO); while MSCOCO is "large" in the number of objects, it has only a small number of categories.

Several issues were raised in the initial reviews regarding the generality of the method and setting of the transition point for switching models. The authors provided detailed feedback and results on a wider set of detector models. One criticism was that the original paper considered CNN-based detectors only. In rebut, authors provided results for some transformer-based detectors also though the reviewers are not completely satisfied. Authors also provided additional details that seemed to satisfy various concerns. After rebut discussion, three reviewers recommended acceptance but one continued to rate the paper as a borderline reject.

The ACs feel that the paper is weak on analysis of detectors such as DINO and the results are justified only empirically but, still, it offers an approach that is worth exposing to the research community. Hence, we recommend that it be accepted. The authors should include the additional analysis and results presented in the rebut but are discouraged from including results that are derived after this period.